# A unifying theory for top-heavy ecosystem structure in the ocean

C. Brock Woodson[1], John R. Schramski[1] & Samantha B. Joye[2]

Size generally dictates metabolic requirements, trophic level, and consequently, ecosystem structure, where inefficient energy transfer leads to bottom-heavy ecosystem structure and biomass decreases as individual size (or trophic level) increases. However, many animals deviate from simple size-based predictions by either adopting generalist predatory behavior, or feeding lower in the trophic web than predicted from their size. Here we show that generalist predatory behavior and lower trophic feeding at large body size increase overall biomass and shift ecosystems from a bottom-heavy pyramid to a top-heavy hourglass shape, with the most biomass accounted for by the largest animals. These effects could be especially dramatic in the ocean, where primary producers are the smallest components of the ecosystem. This approach makes it possible to explore and predict, in the past and in the future, the structure of ocean ecosystems without biomass extraction and other impacts.

[1] School of Environmental, Civil, and Environmental Engineering, University of Georgia, Athens, GA 30602, USA. [2] Department of Marine Sciences, University of Georgia, Athens, GA 30602, USA. Correspondence and requests for materials should be addressed to C.B.W. (email: bwoodson@uga.edu)

Size is considered one of the most important determinants of trophic position and ecosystem structure[1–4]. Metabolic requirements, density, prey preference, prey search capabilities, growth, and reproductive capacity are all related to size for both plants and animals[5–11]. In marine systems, plants (primary producers) are generally the smallest components of an ecosystem. Consequently predation, and therefore trophic level, is almost exclusively size-based. Size-based predation leads to well-defined ecosystem structure in a linear size spectrum, where biomass decreases with size due to inefficient energy transfer[4,7,12–14]. However, many animals either feed across a wide range of prey sizes (large generalist predators, e.g., bears, sharks), or feed lower in the trophic web than expected based on their size alone (mega-consumers, e.g., buffalo, elephants, whales)[15–19]. Both of these feeding modes may directly and indirectly affect the biomass of both the consumers and their resources. Here, using a simple model, we show how the inclusion of large generalist predators and gigantic secondary consumers could substantially increase total biomass and reduces mean trophic level. In marine ecosystems, these animals can invert the trophic structure to a hyperboloid (hourglass) rather than a traditional pyramid shape as biomass is more concentrated in large animals[20–22]. Our results agree with observations in pristine marine ecosystems and provide new perspectives on baselines for how the ocean would look in the absence of over-fishing and other human impacts.

The biomass of organisms across trophic levels is expected to decrease with increasing trophic levels due to inefficient energy transfer and metabolic costs (i.e., the classic trophic pyramid; Fig. 1). When predation is predominantly size-based (e.g., big fish eat smaller fish), a description of ecosystem structure can be derived from the decrease in biomass with individual size called the size spectrum. The slope of the size spectrum ($k$) describes the structure of the ecosystem, where $k < 0$ occurs for bottom-heavy ecosystems with both abundance and biomass concentrated in basal trophic levels (primary producers and consumers). When $k = 0$, biomass is equally distributed across all body sizes, and top-heavy ecosystems occur when $k > 0$. The higher the absolute value of $k$, the more pronounced the shape of the biomass pyramid (Fig. 1). $k$ therefore provides a general quantitative measure of ecosystem structure. For classic trophic pyramids, $k$ is expected to be negative ($k < 0$) and biomass concentrated in lower trophic levels or size classes[7].

However, observations of biomass distributions from pristine coral reefs suggest a fundamentally different pattern, where biomass decreases with trophic level until a significant increase occurs with the largest predators[20,21,23]. Initial estimates of shark populations on Palmyra Atoll were over 100 tonnes km$^{-2}$. Even though these numbers were discounted as significant over-estimates[23], more recent estimates (that claim non-inverted trophic structure) are ∼ ½ these numbers, but still suggest higher biomass of sharks than herbivorous fishes[21]. Explanations for these patterns presently range from overestimation of top

predator biomass[21,23], energetic subsidies[7,24], and the presence of prey refuges[25].

While these explanations could reconcile observations with current theoretical predictions from the size-spectra theory, they do not account for large predators that feed well below the prey sizes (e.g., whales) or on a wider range of prey sizes (e.g., sharks) than typically assumed in size spectra models and theory. For species that feed on a wide range of prey sizes, the mean prey size may increase[17–19], but the median prey size is often invariant[15]. The effects of predators that feed well below prey sizes predicted through size-spectra theory or that feed on a wide range of prey sizes provide an alternative perspective to previous explanations of high top predator biomass in pristine marine ecosystems.

Here we use a thermodynamically balanced, metabolic theory model to show that relaxation of one of the key assumptions in size-spectra theory, namely that animals feed on smaller animals within a confined size range, explains how top-heavy ecosystem structure is possible. The addition of large predators leads to an hourglass-shaped trophic structure with biomass concentrated in the largest animals. Our results compare well with observed trophic structure in pristine marine ecosystems where the biomass of top predators leads to inverted trophic structure. This finding provides an alternate explanation for the empirical observation of top-heavy food webs, and can provide predictions of when and where these ecosystem structures may occur.

## Results

**Size spectra theory.** In size-structured ecosystems, the distribution of biomass with individual size ($k$) is determined by the size of predators relative to their prey, defined as the predator-prey mass ratio (PPMR) and the rate of energy transfer between predator and prey. In our context, PPMR is defined as the mean (for the whole community) ratio of mass at trophic level $n$ divided by the mass at trophic level $n−1$. PPMR > 1 indicates predators larger than their prey, while PPMR < 1 indicates predators smaller than their prey. Energy transfer comprises the effects of metabolism and inefficient energy conversion as well as the energetic costs of foraging and prey capture. All of these biological processes are generally accounted for using an assimilation or trophic transfer efficiency (TE). TE is generally assumed to be ∼ 10%, but may vary widely based on indirect evidence (2.5–40%)[26–28]. The theoretical size-spectra biomass scaling coefficient $k_t$ is related to PPMR and TE as[2,22]:

$$k_t = 0.25 + \log(TE)/\log(PPMR), \qquad (1)$$

where the intercept (0.25) arises from allometric scaling laws that suggest biological function scales as individual mass to the ¼ power[1]. Since the laws of thermodynamics require TE to be less than one, this relationship requires the log ratio be negative unless predators are smaller than their prey[22]. Estimated ranges of PPMR ($10^2$–$10^3$) and TE ($0.101 \pm 0.058$) for marine systems

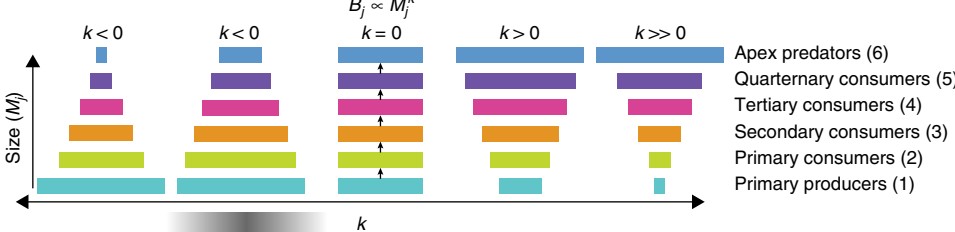

**Fig. 1** Relationship between ecosystem structure and size-spectrum scaling. Width of bars represent relative abundance or biomass for value of the mass scaling exponent, $k$. Gray shading based on the distributions of $k$ for marine ecosystems based on empirical estimates of trophic efficiency (TE) and predator:prey mass ratios (PPMR). Arrows denote flow of energy and hold for all pyramids. Trophic compartment (level) is given on right

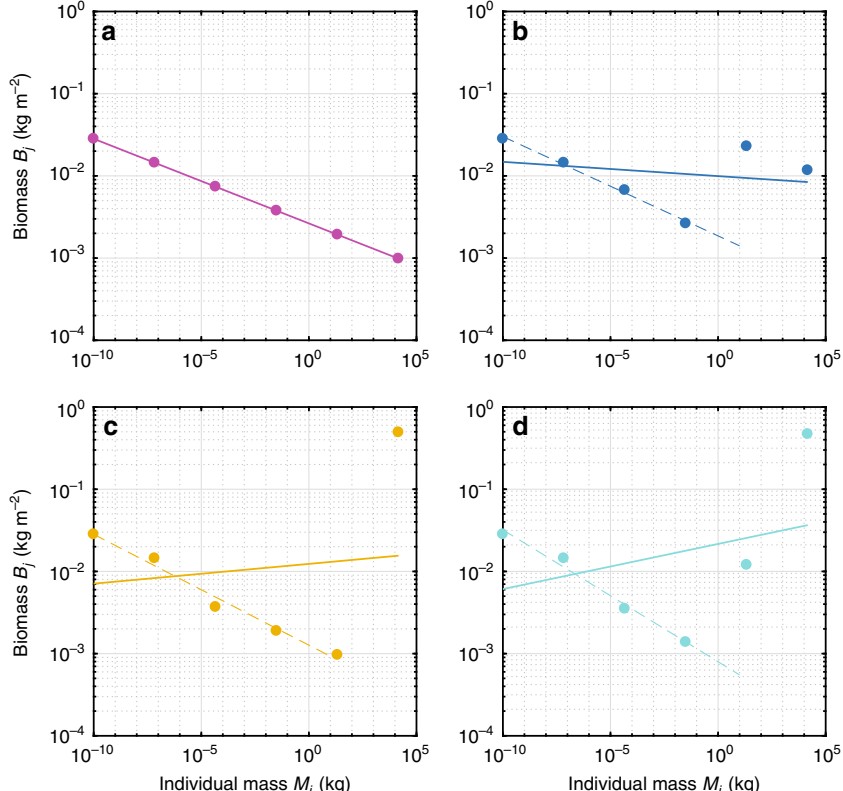

**Fig. 2** Biomass distribution across trophic compartments for size-structured food webs. Biomass versus individual organism size for **a** base case, **b** LGPs, **c** GSCs, and **d** LGPs + GSCs. Dashed line in each plot represents the slope of the biomass spectrum including only trophic compartments up to the fish size range (100 kg)

constrain $k < 0$ (−0.16 to −0.08), insuring bottom-heavy ecosystem structure (Fig. 1; gray shading)[7,27,29]. Observations of size-spectra from individual reefs to entire seas in the size range of marine fishes confirm these expectations with estimates of $k$ from −0.12 to −0.04[2,30]. However, size-spectra theory currently does not account for large generalist predators defined here as animals that feed across a wide range of prey sizes, or animals that feed lower in the food web than predicted from their size (Supplementary Table 1)[31]. For these species, simple averaging of PPMR and TE will not accurately represent energy transfer because prey size is more appropriately represented as a median, and averaging juvenile salmon and whales (both dominant consumers of a single prey, krill) into a single trophic level is not a suitable representation of ecosystem structure.

**Feeding beyond PPMR**. Many large animals (bears, sharks) feed over a wide range of prey sizes and here we call these large generalist predators (LGPs). PPMR for these animals also can be as high as $10^4$ based on estimates of diet composition (Supplementary Table 1). Similarly, some animals (buffalo, elephants, baleen whales, mobulid rays, whale sharks) feed much lower in the trophic web than predicted by size alone. We label these animals mega-consumers or gigantic secondary consumers (GSCs) because they generally feed on primary producers or primary consumers (zooplankton in marine systems) and their PPMR can be as high as $10^6$–$10^8$ (Supplementary Table 1). LGPs and GSCs do not follow size-based predictions for predation leading to food webs that do not follow allometric scaling[3]; as such, other traits have been invoked to explain food web structure[32]. However, the effects of LGPs and GSCs on ecosystem structure and size-spectrum theory have not been addressed[31].

One issue that arises when comparing size spectra and biomass distributions based on trophic level is where to place LGPs and GSCs since they feed on a lower effective trophic level than predicted by the size of the animal. Apex predators should have a trophic level of 5 or 6. However, when including consumption of small prey as LGPs, the trophic level can be reduced to between 4 and 5[33]. Similarly, whales (or GSCs) are closer to a trophic level of 3 rather than 6 as expected for a large animal at the top of the food web. We propose that for assessing biomass distributions, size spectra may more appropriate than trophic level for comparing across ecosystems where mean trophic levels vary considerably. Adjusting biomass distributions to match size-spectra theory unifies these two viewpoints and clearly defines the trophic position of LGPs and GSCs.

We examine the effects of LGPs and GSCs on the distribution of biomass in a theoretical ecosystem using a steady-state compartment ecosystem model based on metabolic theory (Supplementary Tables 2–5). We compare ecosystem characteristics such as biomass, structure ($k$), trophic level, and community mean trophic transfer efficiency (cTE) between size-structured food webs with and without LGPs and GSCs across a range of individual TEs including constant, random, and inversely proportional to individual size ($M_{ind}$)[27].

The distribution of biomass across trophic compartments shows a clear shift with the addition of LGPs and GSCs (Fig. 2). Considerably more biomass is found within upper trophic compartments when accounting for these prey preference behaviors (Fig. 2, Table 1). In most cases, the biomass of LGPs and GSCs exceeds the biomass of primary and secondary consumers leading to positive values of $k$ and top-heavy biomass structure (inverted biomass pyramids). Within the size range of fishes, our results are consistent with size-spectra models that

**Table 1 Mean ecosystem properties for all four trophic webs over 5000 simulations**

| Trophic web | PPMR ($\times 10^4$) | Trophic level | Total biomass (kg m$^{-2}$) | $k$ |
|---|---|---|---|---|
| Base | 0.07 | 6 | 0.06 | −0.08 |
| LGP | 0.16 | 5.5 | 0.10 (+67%) | −0.02 (+73%) |
| GSC | 0.96 | 5 | 0.16 (+167%) | 0.02 (−25%) |
| LGP+GSC | 2.1 | 4.5 | 0.19 (+267%) | 0.06 (+50%) |

Percent increase in parentheses represents the increase relative to the base for biomass and relative to theoretical prediction for $k$. Constant trophic efficiency (cTE = 0.101 ± 0.058)

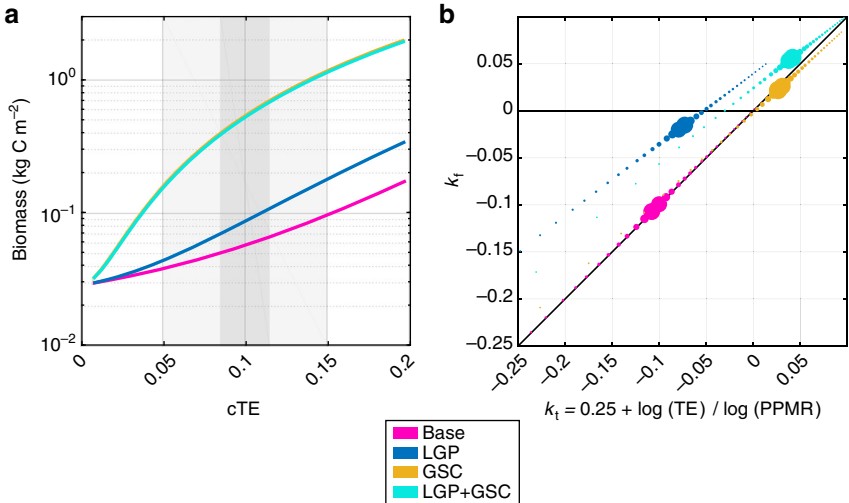

**Fig. 3** Effects of GSCs and LGPs on biomass density and ecosystem structure. **a** Biomass density versus community trophic efficiency (cTE) where gray shading shows distribution of cTE for marine ecosystems (cTE = 0.101 ± 0.058). **b** Mass scaling coefficient from least squares fit, $k_f$, versus calculated theoretical value, $k_t$, from size-spectra theory. Size of data points represents the distribution of cTE for marine ecosystems

focus only on these classes (dashed lines in Fig. 2). The spread of data around the fit increases and becomes non-random when incorporating LGPs and GSCs indicating an overall reduction in the ability of size-spectrums to adequately represent ecosystem structure. However, a general trend emerges with decreasing biomass from primary producers to tertiary consumers, then a dramatic increase in biomass for LGPs and GSCs. This distribution resembles a top-heavy hourglass shape (Fig. 2). Our results are important because they indicate a release from size-based constraints of ecosystem structure due to LGPs and GSCs, in the absence of energetic subsidies[7]. The net effect of both LGPs and GSCs is to shorten the trophic chain (effectively reducing the community trophic level) and increase biomass in large consumers (Table 1).

The addition of LGPs and GSCs creates a complex ecosystem structure where biomass declines from herbivores to planktivores to carnivores, but then dramatically increases for LGPs and GSCs. Such patterns are consistent with recent observations in pristine coral refs.[20,21,23] where the distribution of biomass assumes a top-heavy hourglass shape with the highest biomass in the largest animals regardless of the trophic level (Fig. 2).

Total biomass within an ecosystem is positively proportional to the community trophic transfer efficiency (cTE). However, the presence of both LGPs and GSCs increases total biomass (Fig. 3a; Table 1). The addition of LGPs alone increases biomass by 67% and increases $k$ by 50% above theoretical predictions. GSCs increase biomass by 167%, but $k$ is slightly lower than theoretical predictions. The combination of LGPs and GSCs increases total ecosystem biomass by 267% and $k$ by 50% compared to

theoretical predictions. The additional biomass in each of these scenarios is exclusively stored in large animals (Fig. 2).

Size-spectra models typically follow individual size where scaling arguments for prey capture, prey clearance, and growth rate allow for predictions of ecosystem structure[5,31,34]. In contrast, our model is based on trophic compartments and uses metabolic scaling and thermodynamic relationships to predict ecosystem properties. Yet, both models predict similar scaling for size-structured ecosystems with linear energy transfer between trophic compartments (Fig. 3b). However, the addition of LGPs and GSCs increases the biomass scaling coefficient $k$ above size-spectra theoretical predictions consistent with the observed increase in ecosystem biomass. The addition of LGPs acts more strongly on the scaling coefficient where theoretical estimates of $k$ under-predict the actual estimates from least squares regression. In contrast, GSCs increase both biomass and $k$ but in accordance with size-spectra theory (Fig. 3b). The effects of GSCs and LGPs are synergistic when both are combined, leading to a predominance of top-heavy ecosystems.

Our results are sensitive to changes in TE across trophic compartments. Decreasing TE with organism size therefore may offset any benefits of increased PPMR due to LGPs and GSCs[27]. We ran the model with TE as a function of individual size where TE decreased from 0.30 to 0.03 across the range of animal sizes[27]. However, when accounting for changes in TE with individual mass, identical patterns emerge with both increased biomass and positive scaling coefficients (Fig. 4). Consistent with results using constant TE, LGPs have stronger effects on $k$, whereas GSCs strongly affect both total biomass and $k$. Our results are also

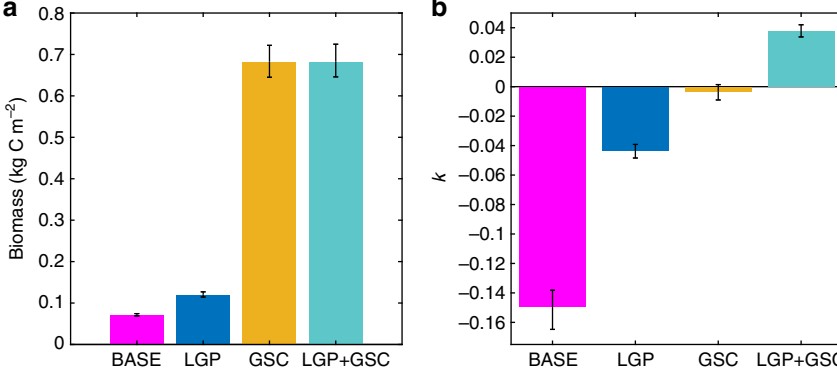

**Fig. 4** Mean biomass and scaling coefficient for ecosystems with TE as a function of size. **a** Biomass and **b** scaling coefficient ($k$) with size dependence of TE based on Eqn (4). Error bars show 95% confidence intervals ($n = 5000$ for all 4 cases)

potentially sensitive to the number of trophic compartments. However, preliminary analyses with 3–10 trophic compartments suggest that hourglass or inverted trophic structure is insensitive to the number of trophic compartments as long as the diet matrix relationships are preserved (Supplementary Fig. 1). Addition of a true apex predator that preys on GSCs and LGPs also does not significantly alter our results in relation to biomass distribution, although the total biomass of apex predators also increases (Supplementary Table 6 and Supplementary Fig. 2). Our results are therefore robust with respect to uncertainty in TE, the number of trophic compartments, and the presence of apex predators; however these sensitivities will still be an important avenue of future work.

Inverted or top-heavy trophic structure is expected when the *PPMR* is greater than $10^4$, or when $\log_{10}(\mathrm{TE})/\log_{10}(\mathrm{PPMR}) > -0.25$ ($\mathrm{PPMR}^{-0.25} < \mathrm{TE}$)[22]. For marine trophic interactions in the fish size range, PPMR ranges between 200 and 3000, and TE ~ 0.10 yielding $k < 0$. However, incorporating LGPs and GSCs, the effective community PPMR can increase significantly to greater than $10^4$ (Table 1), leading to $k > 0$ and top heavy trophic structure. A similar theoretical consideration is that inverted trophic structure occurs when the ratio of the growth rates of predator to prey are greater than the trophic transfer efficiency. This formulation is identical to the constraint derived through metabolic scaling with individual growth rate given by $g = P_o M^{\beta-1} e^{-(E/kT)}$ leading to positive biomass scaling when $g_{\mathrm{predator}}/g_{\mathrm{prey}} = (M_{\mathrm{predator}}/M_{\mathrm{prey}})^{-0.25} = \mathrm{PPMR}^{-0.25} > \mathrm{TE}$.

Incorporating LGPs and GSCs into metabolic scaling and size-spectra theory has two surprising effects, increased total community biomass and top-heavy ecosystem structure. Both are achieved due to a lower effective trophic level and higher PPMR consistent with both size spectra theory and metabolic scaling[1,7,34,35]. The observed hourglass shaped distribution of biomass emerges when at least 4% of energy allocated to LGPs is derived from the smallest prey, and < 1% of available production (e.g., 0.1% of total production) of zooplankton is allocated to GSCs, so should be expected across a wide range of ecosystems (Supplementary Figs. 3 and 4).

## Discussion

We show that top-heavy trophic structure in marine ecosystems is theoretically possible based on energetic balance and metabolic scaling. The biomass of LGPs and GSCs can be significantly larger than that from lower trophic levels, but the shape of the biomass distribution is more likely hourglass-shaped rather than a pyramid. Our results are consistent with both size-spectra theory and observations of top-heavy biomass structure in pristine marine ecosystems; thus, providing a unifying mechanism to understand

ecosystem structure. However, there are a few complexities that must be considered when assessing biomass distributions.

Traditional biomass pyramids should not always be expected in actual ecosystems. Our results do not indicate an inverted biomass pyramid, but rather a top heavy hourglass distribution of biomass when considering the biomass pyramid strictly from a size-based lens (e.g., whales are the top of the pyramid even though their true trophic level is much lower). However, even when considering only LGPs, the top-heavy hourglass shape is the predominant distribution regardless of how trophic efficiency varies among trophic compartments. The top-heavy hourglass shape is similar to observations in pristine marine ecosystems even after corrections for overestimation of LGPs[11,13] and using a simplified coral reef food web (Fig. 5; Supplementary Table 6). Therefore, we expect that irregularly shaped biomass distributions are the norm due to the complexity of predatory behaviors in the ocean, but that size-spectrum based estimates will hold reasonably well for animals in the size range of fishes[12,31].

Our model also assumes steady-state conditions. Ecosystems are complex and dynamic and may never approach a true steady state, and consequently our results suggest a maximal carrying capacity, not necessarily a realized pattern. Simply put, top-heavy trophic structure is possible energetically. However, the similarity of our results to observed patterns in pristine coral reefs (Fig. 5) suggest that either the undisturbed ocean was near steady state, or that the distribution of biomass maintains the patterns, but not the magnitude of theoretical predictions.

Mass-balanced, dynamic models of ecosystems do not give rise to inverted hourglass structure[36]. However, the ability of these models to represent fine-scale predator-prey interactions may underestimate their predictive power[37,38]. Production at higher trophic levels could be as much as 10–100 times higher than predicted in traditional mass-balance models due to fine-scale aggregation of prey[37,38]. Another explanation could be that the carrying capacity in mass-balanced models is greatly underestimated (as suggested by our model results). Increased production or carrying capacity would likely lead to inverted hourglass trophic structure in current dynamic ecosystem models used to estimate fisheries production and health of global fisheries[36,39] by allowing more biomass to accumulate at higher trophic levels.

Estimates of TE remain very uncertain and could have large effects on our model results especially if TE is a stronger function of size than presently believed[27]. However, we tested the model across a wide range of TE randomly distributed from 1 to 50%. Our results were robust to this variation in TE suggesting that these patterns are consistent, but that the total biomass in the system may change significantly depending on the community mean trophic level (cTE). In addition, our results can be used to estimate TE directly.

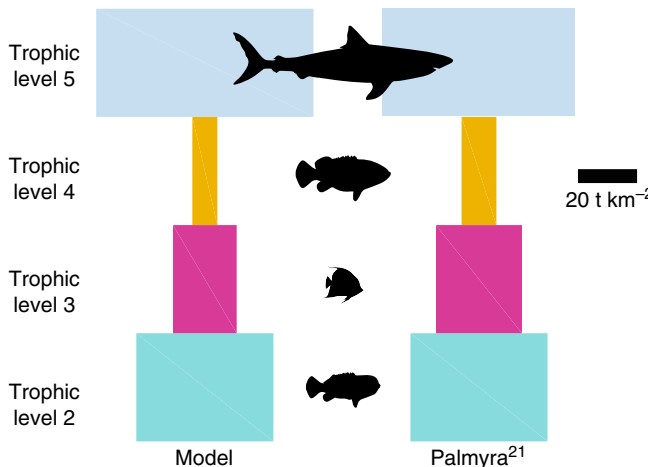

Trophic level 5

Trophic level 4

20 t km$^{-2}$

Trophic level 3

Trophic level 2

Model          Palmyra[21]

**Fig. 5** Comparison of ecosystem structure shape between model and results from Palmyra Atoll. Palmyra data reproduced from refs. [20, 21]. Width of each rectangle shows the relative biomass of each trophic level

For example, using our model results with a simplified coral reef food web (Supplementary Table 7) and comparing to results from the Palmyra Atoll, we can indirectly estimate TE by fitting the estimated biomass at each compartment and using the TE to adjust and appropriately scale the biomass estimate from the model (Fig. 5). Interestingly, this scaling suggests decreasing trophic efficiency with size from 0.23 for herbivorous fishes to 0.09 for sharks with a cTE of 0.13. Our estimate of cTE is in agreement with general assumptions about TE in marine systems globally where TE decreases with animal size[27] and with a mean around 10%. A decrease in TE with animal size is largely believed to be the result of increased foraging effort, which may be counteracted when accounting for fine-scale prey aggregation[37]. Further inquiry into TE is warranted but beyond the scope of the data presented here.

Our model will require significant ground-truthing before being applicable generally to understanding marine ecosystem structure. However, from the simplified coral reef model above, we can also estimate the minimum production required to support populations of sharks and other top predators. Based on our model, production at Palmyra Atoll needs to be at least 0.47 kg C m$^{-2}$ yr$^{-1}$ to support populations reported by Sandin et al.[20] and at least 0.21 kg C m$^{-2}$ yr$^{-1}$ to support more recent population estimates reported by Bradley et al.[21]. The annual mean production for Palmyra, based on 10 years of satellite-derived primary production data, is 0.29 ± 0.04 kg C m$^{-2}$ yr$^{-1}$, and reef production can be up to 73% higher than satellite-based estimates of surface production[40]. Hence, the populations reported by Sandin et al. could even be supported within our uncertainty depending on the proportion of small prey consumed by sharks or TE.

An alternative, yet similar view is to estimate the area required to support a viable population of top predators. Building on our example from Palmyra and assuming this population is viable at ~ 2000 gray reef sharks[21], we can estimate the area of reef required to support this population. Using production values above (0.29 ± 0.04 kg C m$^{-2}$ yr$^{-1}$) and our model, supporting a population of 2000 gray reef sharks would require ~ 17 km$^2$, well below the area of the reef surrounding Palmyra. The area needed to support all top predator biomass (65–145 tonnes km$^{-2}$) would be ~ 46–103 km$^2$. The area of the Palmyra reef is ~ 80 km$^2$. Our results suggest that (1) populations of top predators on Palmyra can be locally supported, (2) these populations are at or very near local carrying capacity, and (3) the natural trophic structure of pristine coral reefs is an inverted hourglass shape.

Finally, estimates of carrying capacity and trophic efficiency used in population dynamics models for LGPs and GSCs may be vastly underestimated[36]. As both carrying capacity and TE limit population growth of all animals in a population dynamics model, it is likely that these terms need to be re-evaluated using metabolic scaling to improve ecosystem models, especially those used in conservation and fisheries management contexts[36,41].

Our results indicate that top-heavy trophic structure may be possible in marine ecosystems when accounting for large generalist predators or gigantic secondary consumers without invoking energetic subsidies, overestimation, or prey refuges[7,21,23,24]. Such trophic structure may also be possible in terrestrial systems with large animals (elephants) that feed low on the food chain such as in sub-Saharan Africa[42]. Our results are robust to variation in TE and suggest that top-heavy hourglass shaped biomass distributions are likely the norm for undisturbed marine ecosystems[21]. Recent observations of bottom-heavy trophic structure consequently may be the result of anthropogenic defaunation[43–45].

Our model provides an alternative complimentary approach for generating baseline expectations of what ecosystem structure would be without extraction and other impacts (e.g., setting a baseline and target for assessing recovery in MPAs or other conservation actions). Questions still remain however regarding the total biomass in undisturbed ecosystems, how more complex, realistic dynamics food webs might influence trophic structure[31], and finally how top-heavy biomass distributions may change our views of human impacts on the ocean, especially marine defaunation[43,46].

## Methods

**Model development**. We developed a steady-state metabolic theory-based compartment model to examine the role of LGPs and GSCs on ecosystem biomass and structure. The model estimates steady-state biomass for a suite of trophic compartments using metabolic theory. For each trophic compartment, mean individual mass ($M_{ind}$) is specified. Individual production is computed from metabolic scaling as[35]:

$$P_{ind} = P_o M_{ind}^{\beta} e^{-E/kT} \qquad (2)$$

where $\beta$ is the quarter-power allometric mass scaling coefficient (3/4), $E$ is the activation energy (0.32 eV for phytoplankton, 0.65 eV for consumers), $k$ is Boltzmann's constant (8.62×10$^{-5}$ eV K$^{-1}$), and $T$ is temperature in Kelvin. $T$ is set to the environmental temperature for ectotherms (287 K) and internal temperature for endotherms (310 K). At steady state, the production for each trophic compartment ($P_j$ where $j$ subscript refers to trophic compartment) is the sum of the production by prey ($P_{j-1}$) accounting for inefficient energy transfer through the trophic transfer efficiency (TE):

$$P_j = TE \sum P_{j-1} = N_j P_{ind(j)} \qquad (3)$$

where $N_j$ is the number of individuals within trophic compartment $j$, and the index $j-1$ refers to all prey for $j$. The biomass ($B_j$) of each trophic compartment is then $N_j M_{ind}$. Once $N_j$ and $B_j$ are computed, ecosystem structure and biomass characteristics can be examined.

Flow between trophic compartments can be simulated in several fashions. In our model, we first consider a size-spectrum where energy is transferred from one trophic compartment to the next larger size compartment (Supplementary Table 2). We used a model with 6 trophic compartments with individual masses distributed logarithmically between 10$^{-9}$ and 10$^4$ kg. This formulation is theoretically identical to size-spectrum models that are common in fisheries literature as will be demonstrated later.

To address the effects of LGPs on ecosystem biomass and structure, we changed the trophic web so that the second largest trophic compartment acquires energy across the three previous compartments weighted towards the closest (Supplementary Table 3). Similarly, we modified the food web to address GSCs by allowing the top compartment to prey on zooplankton (Supplementary Table 4), or three trophic levels below expected in a size-structured community. Finally, we incorporated both GSCs and LGPs in a single ecosystem to assess the combined effects of these groups (Supplementary Table 5).

**Simulations**. We ran 1 million simulations (250,000 for each scenario) where TE was randomly assigned a value for each trophic compartment between 1 and 50%

for each run. We found that convergence of results occurred around 2000 simulations (Supplementary Fig. 5). We therefore ran other model scenarios with 5000 simulations each (20,000 total simulations) to ensure statistical convergence of our results.

We also ran 5000 simulations with TE dependent on organism size[27] based as:

$$\text{TE} = 0.16 M_{\text{ind}}^{-0.07} \tag{4}$$

Using (4), TE varies from 0.3 for the smallest trophic compartment to 0.03 for the largest. For each simulation, the trophic efficiency (TE), individual mass ($M_{\text{ind}}$), biomass ($B$), and abundance ($N$) of each trophic compartment is saved. From these data, we can compute the total biomass for the ecosystem, PPMR, and the abundance and biomass scaling coefficients ($k$).

We computed the PPMR for each trophic compartment using a weighted average based on the energy flow from each compartment. This resulted in the PPMR = $M_{\text{predator}}/M_{\text{prey}}$ for all but large generalist predators. We then calculated community mean PPMR (cPPMR), trophic efficiency (cTE), trophic level (cTL), and the slope of the biomass spectrum, $k_{\text{f}}$. PPMR and cTE were computed as the biomass weighted geometric mean for each value as:

$$\log(\text{cPPMR}) = \frac{1}{\log(B_{\text{tot}})} \sum_j \log(\text{PPMR}_j B_j)$$

We estimated $k_{\text{f}}$ for each simulation using a least-squares regression. We explored estimation of $k_{\text{f}}$ using more advanced Bayesian regression however the results were virtually identical so we used traditional least-squares regression for these estimates.

**Sensitivity analyses**. To test the sensitivity of the model to our results, we further ran > 40,000 simulations where we specified the proportion of energy for both LGP and GSC diet in increments of 0.05 between small, medium, and large prey. Large prey are the prey item that matches size-spectra theory. Results of the sensitivity analyses for diet matrices, proportional energy, and number of trophic compartments are presented in Supplementary Figs. 1–4, respectively. We used these results to estimate the minimum energy required for inverted pyramids by evaluating the energy needed for the biomass of the LGP to be greater than the biomass of large prey.

**PPMR estimation**. To estimate PPMR and classify LGPs and GSCs, we used diet preferences reported in the literature[16,33,47,48] and calculated the mean and median prey size using common masses for predators and prey reported in FishBase (www.fishbase.org). For ~ 100 fishes examined, mean and median prey size were statistically similar yielding PPMRs ~ $10^2$. Large differences occurred in these values for sharks, tunas, baleen whales, and mobulid rays (Supplementary Table 2). Therefore, we defined LGPs as predators where median prey size was an order of magnitude smaller than mean prey size, and GSCs as predators where PPMR is greater than $10^6$.

**Code availability**. Model code used for trophic structure calculations is provided in Supplementary Note 1. All model code (Python) including figure scripts (Matlab) used in this study is available through github at https://github.com/cbrockw/ecosystem_structure.git.

**Data availability**. The data sets generated and analyzed during the current study are available from the corresponding author on reasonable request. Model results are publicly available through the Gulf of Mexico Research Initiative Information & Data Cooperative (GRIIDC) at https://data.gulfresearchinitiative.org (10.7266/N7959G1K).

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

## Acknowledgements

We like to thank J.H. Brown and F. Micheli for helpful comments on earlier drafts, A.T. Greer, S.Y. Litvin, and M. Pinsky for helpful discussions during early phases of the study. C.B.W. was supported by NSF OCE-1212124, 1416837, and 1536618. This work was made possible in part by a grant from The Gulf of Mexico Research Initiative supporting the Ecosystem Impacts of Oil and Gas in the Gulf research consortium (to S.B.J.). This is ECOGIG contribution 492.

## Author contributions

C.B.W. and J.R.S. conceived the project, designed the study, and analyzed data. C.B.W. wrote code and performed simulations. C.B.W., J.R.S., and S.B.J. wrote the primary manuscript. All authors approved final manuscript.

## Additional information

**Competing interests:** The authors declare no competing financial interests.

