## [Peer Review File · Nature Communications]

Reviewers' comments:

Reviewer #1 (Remarks to the Author):

The paper claims to provide a "unified theory" for how top-heavy biomass distributions (i.e. where total biomass in the ecosystem is greatest at largest body sizes) can arise in marine ecosystems, based on the presence of large consumers that "feed down food chains". This is a novel approach that seeks to directly address the need identified by other authors to develop new MTE-based models that extend the "energetic equivalence with trophic transfer correction" macroecological framework to incorporate more diverse foraging strategies than the overly simplistic "big fish eat little fish" paradigm. The authors should be commended for seeking to forge new ground in this direction. However, in its current form the manuscript is unfortunately hamstrung by "oversell", a related problem of skewed framing in relation to the literature, and glossed-over and ill-defined assumptions. I do feel that there is the nucleus of a novel and important contribution here. However I also think that much more careful consideration of the assumptions and potential flaws of the proposed model is required, along with a more balanced and considered delivery.

Oversell & framing in relation to literature:

- throughout the manuscript the hypothetical nature of the model evidence is obscured by overly definitive language.
- its not at all clear why previous explanations of top-heavy biomass distributions are dismissed as not providing a "fundamental mechanism" or "theoretical bridge" between observations and scaling theory, while this manuscript apparently does. It would seem more balanced to frame this as an additional perspective/new hypothesis (models are, after all, hypotheses).
- the framing in relation to previous studies of the structure of coral reef communities is strange. For example, line 112-113 states that the predicted strongly inverted structures are "consistent with recent observations in pristine coral reefs" and supports this statement by citing 4 papers (Nadon et al., Sandin et al., Bradley et al., and Mourier et al.). However of these papers only Sandin et al supports strongly inverted locally inverted structures. Both Nadon and Bradely suggest that predator biomass was substantially over-estimated by Sandin et al., and Mourier documents a breeding aggregation (i.e. community structure supported by external subsidies).

major/general comments

- my most substantive concern with the approach and assumptions is that, so far as I can tell, the addition of LGPs and GSCs seems to be "double dipping". A TE of 10% should imply that 10% of production in a compartment is propagated to higher trophic levels. Hence if 5% is consumed by LGPs and GSCs, that only leaves 5% for the next largest compartment for a total TE of 10%. This implies that there should be a compensatory relationship between TE for LGPs and GSCs and lower trophic level compartments. As far as I can tell, this is not the case.
- the sensitivity of the results to both the number of size classes (compartments) and the parameterization of the interaction matrices is not considered. This is particularly concerning given that there seem to be disparities between the approximate prey mass preferences listed in table S1 and the interaction coefficients in tables S2-5 (e.g. GSCs consume the 10^{-4} kg "zooplankton" compartment, which would be an order of magnitude smaller than the smallest prey preference - for manta rays - listed in table 1)
- another concern is that there is no consideration of how much total production is required to sustain minimum viable population sizes of the largest consumers, and how this affects interpretation for real communities. This ties back to the concept of why large consumers may be forced to adopt strategies of feeding down food chains to begin with, as noted by previous authors.
- other modelling approaches other than size spectra models re-distribute production across compartments and include LGPs and GSCs (or equivalent functional groups), but do not give rise to

the same top-heavy configuration. It would be informative to discuss the reason for this disparity.

- the reproducibility of this approach, and ability of readers to satisfy concerns regarding model structure and assumptions, would be greatly enhanced by providing code as part of the supplementary materials.

specific comments:

- lines 4-7: this is a really difficult to digest sentence, and would benefit from a reword (probably breaking into several sentences)
- line 9: its not clear that sharks (or bears for that matter) feed on a larger range of prey sizes *relative to their body size* than do smaller consumers, and no citations are provided to support this assertion. Referring to some of the extensive literature on predator-prey size relations would be worthwhile (by folks such as Gabriel Costa, Marlee Tucker, Frank Scharf & Francis Juanes)
- throughout the introductory part of the MS it would be helpful to more carefully distinguish where you are referring to biomass vs. abundance spectra (and their slopes)
- line 46: all of the studies cited here provide indirect evidence that TE may be more variable than widely assumed. I suggest a minor reword to reflec this.
- line 50: as written this suggests that TE nay be >1 if preys are smaller than prey, which is misleading.
- lines 81-83: its not clear what is meant here. Suggest rewording to clarify.
- tables S1-5: it seems strange that apex predators are assigned to the same size class as GSCs

Rowan Trebilco

Reviewer #2 (Remarks to the Author):

Review of "A unifying theory for top-heavy ecosystem structure in the ocean"

This is a very intriguing piece of work. How to include of large, mobile, generalist consumers in food web theory is one of the most fascinating aspects of community ecology. This paper focuses on the size spectrum, and provides a new way to integrate unusually large and generalist predators into size-spectral theory. Basically, the author(s) demonstrate that the inclusion of these predators result in feasible top-heavy hourglass ecosystems, without invoking energetic subsidies, overestimation, or prey refuges, as done previously. They do so by means of a steady-state model using metabolic theory reasoning across trophic compartments, and illustrate their findings with examples from marine ecosystems (coral reefs mostly). But their model can be applicable to terrestrial systems.

While I found the question fascinating, and I really like the generality of the approach taken, I have two major concerns with this manuscript that, unless they are solved, I cannot fully evaluate the validity and potential impact of this paper.

First, large generalist predators and mega-consumers increase the total biomass and reduce the mean trophic level in an ecosystem- in particular, in marine ecosystems. The authors claim that size-spectral theory does not account for generalist predators or animals that feed lower in the food web than predicted from size alone. I do not see why that is the case. In equation 1, TE and PPMR can be calculated for any feeding structure by simply averaging effects. The $\frac{1}{4}$ exponent, however, is not clear to me whether it assumes feeding on a single resource pool (e.g., a single trophic level). Unless the author(s) explained more clearly what the basis and the gap in the theory comes from, it is very difficult to evaluate the novelty and potential impact of this paper in the area.

Second, the author(s) state that the spatial scale over which size-spectra are calculated corresponds

to the range of the largest or most mobile animal in the system. Also, that the existence of energetic subsidies for large and generalist predators is not well integrated into the size-spectral theory. With their toy model, the authors show that there is no need to claim the existence of prey refugia or energetic subsidies. This is novel and counter-intuitive. However, it remains to be proven that the spatial scale at which these large and generalist predators feed is the one contemplated in size-spectral theory, or in the model used here. Some of these predators have migration patterns that would require building size spectra at large spatial extents. The question that remains is: would current theory still work if size-spectra were built at the scale at which these large and generalist predators actually operate? If so, do we still need the theory developed here?

Minor points:

- Manuscript is well written but the lack of subheadings makes it difficult to navigate and focus attention. I strongly suggest using key subheadings.
- I found the Intro very repetitive- lines 1-74 can be shortened to half their current extent.
- Use of abundance and biomass as interchangeable terms? If by abundance, the author(s) refer to numerical abundance, predictions are very different. For example, Figure 1 only makes sense for biomass, not for numerical abundance. Please use only biomass throughout the manuscript.
- Lines 14-16: empirical evidence for this should be cited.
- Line 23: references needed
- Figure 1: grey shading (based on the distributions of k for marine ecosystems)
- Line 48: Where equation 1 is derived from? This is the key theoretical construct of the manuscript, yet it seems to emerge from nowhere.
- Lines 150-152: This is a very interesting and convincing argument. However, the authors should do a better job in explaining the equations.
- Line 161: should read "larger than that from lower..."
- Line 197-199: references are needed here. It is not clear to me why this should be the case. Based on stoichiometric reasoning, animal-animal interactions should have larger efficiencies than herbivore-phyto interactions- hence efficiencies should increase with size.
- Supplementary Table 1: please explain what abbreviations mean

Reviewer #3 (Remarks to the Author):

The authors present a model that predicts the size spectrum of a community that contains both large generalist predators (LGP) and gigantic secondary consumers (GSC), both of which are not well-predicted by current size spectrum theory. The model is a trophic compartment model that is based on metabolic theory and steady state assumptions. The authors show that their model predicts an hourglass-shaped size spectrum where for \leq fish size class, there is a decrease in biomass with increasing body size; above the fish size class there is an increase in biomass with increasing body size when LGPs and GSCs are incorporated into the model. This hourglass shape is energetically feasible and is supported by data collected from pristine communities - it appears that heavily anthropogenically altered communities may lose the top part of the hourglass, biasing observations towards a linear (negatively sloped) size spectra.

I thought that this manuscript was well-written, appeared to be theoretically sound, and provided an important link connecting communities with GSCs and LGPs to the larger body of size spectra theory.

General comments:

It is eluded to several times that human-influenced communities may be absent of GSCs and LGPs, such that anthropogenically altered systems will fit well with traditional size spectrum theory.

Mentioned below is a reference that reports similar biomass skews in terrestrial African systems, which are more Pleistocene-like than other terrestrial mammal systems. Incorporating these terrestrial studies quantitatively into the analyses, or spending some time discussing similarities/differences of these systems with the presented marine systems may help to generalize the findings presented here.

Moreover, I felt that the anthropogenic alteration angle was mentioned in the beginning of the manuscript, but not revisited satisfactorily towards the end of the paper.

L113: This is also something that has been recently shown for African mammalian systems, where the biomass of megafaunal herbivores such as elephants appears to account for a large portion of mammalian biomass in the system - see Hempson et al. (Science, 2015). From Hempson: "Elephants dominate African herbivore biomass, often having biomasses equivalent to those of all other species combined.", which seems relevant here.

It seems like the trophic compartment model is minimalist in the sense that it works well to address the primary questions that the authors are tackling. It is also somewhat divorced from other trophic web models in that it does not consider individual species, but aggregates of individuals from many species within a given trophic compartment (if I understand the model correctly). I'm not suggesting that the authors adopt a different approach, as this one appears to work very well, however I think that drawing some links between the trophic compartment models here and species-explicit trophic web frameworks might be informative/interesting, and help to connect this work to some of the size-based food web frameworks that exist.

Of particular relevance is the paper by Rudolf Rohr (cited below), where they model food webs based on predator-prey mass ratios and have additional 'latent traits' that define non-body mass dictated relationships (e.g. baleen whales). I think that the impact of this contribution would be extended by finding the common ground between these two bodies of work, where one does not find much cross-pollination (surprisingly).

Some references in the food web world that deal heavily with allometric relationships:

-Brose, U., Williams, R. J., & Martinez, N. D. (2006). Allometric scaling enhances stability in complex food webs. *Ecology Letters*, 9(11), 1228–1236. <http://doi.org/10.1111/j.1461-0248.2006.00978.x>

-Petchey, O. L., Beckerman, A. P., Riede, J. O., & Warren, P. H. (2008). Size, foraging, and food web structure. *Proceedings of the National Academy of Sciences of the USA*, 105(11), 4191.

-Rohr, R. P., Scherer, H., Kehrl, P., Mazza, C., & Bersier, L.-F. (2010). Modeling food webs: Exploring unexplained structure using latent traits. *American Naturalist*, 176(2), 170–177. <http://doi.org/10.1086/653667>

-Woodward, G., Ebenman, B., Emmerson, M., Montoya, J. M., Olesen, J. M., Valido, A., & Warren, P. H. (2005). Body size in ecological networks. *Trends in Ecology and Evolution*, 20(7), 402–409.

Minor

L42: consider stipulating what $PPMR > 1$ or < 1 means, so that readers will know for certain whether prey or predator mass is in the numerator. Depending on the method, predator-prey mass ratios are not always presented with one of the two always being in the numerator, which can make things confusing if the reader is used to a different standard.

L50: This line is confusing to me. $\log(TE)$ will certainly be negative due to thermodynamics, but the log ratio will only be negative if the predator is larger than the prey. You say this in the sentence, but it is confusingly put. The relation doesn't 'require' the log ratio to be negative because it can easily be positive, i.e. if the predator is smaller than the prey, which is often the case, particularly in terrestrial systems.

Eq1: Consider mentioning the scaling law that determines the intercept - otherwise it just looks a bit random.

Fig 2: The red line in each subplot is a bit confusing... My eye wanted to match it to the red line in (a) where the points are also red. Consider using a same-color, but stippled or dotted, line to show the fit up to fish size, since the points are color coded from (a-d).

Fig. 3a is not referenced in the text.

L233: is it specified that j is trophic level? I think it is, but I'm not sure. Now I see it is, but consider mentioning this when the subscript is first used.

Figures in general: consider putting more descriptive names on the axes rather than just the parameter. Then you don't have to dig through the paper to figure out what is being conveyed.

Response to Reviewer #1

The paper claims to provide a "unified theory" for how top-heavy biomass distributions (i.e. where total biomass in the ecosystem is greatest at largest body sizes) can arise in marine ecosystems, based on the presence of large consumers that "feed down food chains". This is a novel approach that seeks to directly address the need identified by other authors to develop new MTE-based models that extend the "energetic equivalence with trophic transfer correction" macroecological framework to incorporate more diverse foraging strategies than the overly simplistic "big fish eat little fish" paradigm. The authors should be commended for seeking to forge new ground in this direction. However, in its current form the manuscript is unfortunately hamstrung by "oversell", a related problem of skewed framing in relation to the literature, and glossed-over and ill-defined assumptions. I do feel that there is the nucleus of a novel and important contribution here. However I also think that much more careful consideration of the assumptions and potential flaws of the proposed model is required, along with a more balanced and considered delivery.

We thank the reviewer for an insightful and constructive review of our manuscript. These comments and suggestions were very beneficial and helped us improve our paper.

Oversell & framing in relation to literature

- throughout the manuscript the hypothetical nature of the model evidence is obscured by overly definitive language.

These are important observations and we take them seriously. We have toned down the language with the comments in mind and using our judgment. Without specific examples, it is difficult to definitively address the reviewer's concerns. However, given the changes we made and that the other two reviewers did not raise this as an issue, we feel confident that the manuscript has been enhanced in a way that is a balance among all views provided. We have specifically added a sentence to cast our approach as a new perspective/hypothesis (Lines 41-48), as well as an example of empirical evidence that supports our hypothesis (Lines 224-234).

- its not at all clear why previous explanations of top-heavy biomass distributions are dismissed as not providing a "fundamental mechanism" or "theoretical bridge" between observations and scaling theory, while this manuscript apparently does. It would seem more balanced to frame this as an additional perspective/new hypothesis (models are, after all, hypotheses).

Agreed, we reframed as an additional perspective/new hypothesis. While the model is just a hypothesis, evidence does support it. We have specifically added a sentence to cast our approach as a new perspective/hypothesis (Lines 41-48).

- the framing in relation to previous studies of the structure of coral reef communities is strange. For example, line 112-113 states that the predicted strongly inverted structures are "consistent with recent observations in pristine coral reefs" and supports this statement by citing 4 papers (Nadon et al., Sandin et al., Bradley et al., and Mourier et al.). However of these papers only Sandin et al supports strongly inverted locally inverted structures. Both Nadon and Bradley suggest that predator biomass was substantially over-estimated by Sandin et al., and Mourier documents a breeding aggregation (i.e. community structure supported by external subsidies).

We have rearranged this text to more clearly reference appropriate papers (e.g. Sandin for strongly inverted biomass and Nadon & Bradley for lower values, and Mourier for breeding aggregations). However, although Bradley suggests over-estimation by Sandin, the pyramid is still strongly inverted (as

our figure 4), and Sandin and Bradley's results are both within our model estimates (so maybe neither are wrong but that temporal variability is the main factor). This clarity is a very nice improvement and we are grateful (Lines 32-48).

Major/general comments

- my most substantive concern with the approach and assumptions is that, so far as I can tell, the addition of LGPs and GSCs seems to be "double dipping". A TE of 10% should imply that 10% of production in a compartment is propagated to higher trophic levels. Hence if 5% is consumed by LGPs and GSCs, that only leaves 5% for the next largest compartment for a total TE of 10%. This implies that there should be a compensatory relationship between TE for LGPs and GSCs and lower trophic level compartments. As far as I can tell, this is not the case.

Agreed. Fortunately, this is not how the model works. First, 10% of new production is made available to all predators. This energy is allocated to predators based on prey preference. Then TE is applied to each predator's allocation to estimate predator production. This formulation insures that no more than 10% of production is made available for new production at all higher trophic level consumers.

- the sensitivity of the results to both the number of size classes (compartments) and the parameterization of the interaction matrices is not considered. This is particularly concerning given that there seem to be disparities between the approximate prey mass preferences listed in table S1 and the interaction coefficients in tables S2-5 (e.g. GSCs consume the 10^{-4} kg "zooplankton" compartment, which would be an order of magnitude smaller than the smallest prey preference - for manta rays - listed in table 1)

There were errors in the masses used in the models. For all theoretical models, the masses for each compartment are the same and are 10^{-8} , 10^{-4} , 10^{-2} , 10^0 , 10^2 , 10^4 . These errors have been corrected and this resolves the issues between tables S1 and S2-5 as mentioned. We have completed full sensitivity analyses for the interaction matrices. We have added supplementary figures 2 and 3 to show results of the sensitivity analyses to parameterization of the interaction matrices. We do not provide a sensitivity analysis for the number of trophic compartments since all model results are for a fixed number of compartments. However, we have performed these analyses and they will be reported in a subsequent manuscript that looks at variation in complex food webs. We would be willing to include these here, but do not feel they are necessary for our results in this manuscript. Text referring to the sensitivity analyses was also added to methods (Lines 327-333).

- another concern is that there is no consideration of how much total production is required to sustain minimum viable population sizes of the largest consumers, and how this affects interpretation for real communities. This ties back to the concept of why large consumers may be forced to adopt strategies of feeding down food chains to begin with, as noted by previous authors.

We are not sure why this is a concern as our data are all reported in densities. However, in response to the reviewer's comment here, we have added a discussion of the minimum production required to maintain populations of sharks in Palmyra and compare to satellite-derived primary production climatologies (Lines 235-247). We think this discussion provides excellent evidence that top-heavy hourglass structure is possible.

- other modelling approaches other than size spectra models re-distribute production across compartments and include LGPs and GSCs (or equivalent functional groups), but do not give rise to the same top-heavy configuration. It would be informative to discuss the reason for this disparity.

We added a discussion of our logic explaining this disparity, largely the underestimation of production due to prey aggregation (e.g. Woodson & Litvin 2015). This is a nice addition and we are grateful for the comment (Lines 211-216).

- the reproducibility of this approach, and ability of readers to satisfy concerns regarding model structure and assumptions, would be greatly enhanced by providing code as part of the supplementary materials

We have provided the code (subfunction for computation of trophic structure) as part of the supplementary materials as suggested. We will also upload model code to github once paper is published.

Specific comments

- lines 4-7: this is a really difficult to digest sentence, and would benefit from a reword (probably breaking into several sentences)

Done as suggested. The 3 sentences are better. Thanks (Lines 4-8).

- line 9: its not clear that sharks (or bears for that matter) feed on a larger range of prey sizes *relative to their body size* than do smaller consumers, and no citations are provided to support this assertion. Referring to some of the extensive literature on predator-prey size relations would be worthwhile (by folks such as Gabriel Costa, Marlee Tucker, Frank Scharf & Francis Juanes).

Done as suggested. Thanks for pointing these references out. Several do support this statement that as predator size increases, they feed on a larger range of prey sizes. We have added references accordingly (Lines 10, 44).

- throughout the introductory part of the MS it would be helpful to more carefully distinguish where you are referring to biomass vs. abundance spectra (and their slopes)

Done as suggested. We only use biomass spectra throughout now.

- line 46: all of the studies cited here provide indirect evidence that TE may be more variable than widely assumed. I suggest a minor reword to reflect this.

Done as suggested (Line 57).

- line 50: as written this suggests that TE may be >1 if preds are smaller than prey, which is misleading.

Corrected (Lines 61-64).

- lines 81-83: its not clear what is meant here. Suggest rewording to clarify.

Done as suggested (Lines 102-104).

- tables S1-5: it seems strange that apex predators are assigned to the same size class as GSCs

LGPs and GSCs are separate size classes. We have corrected Tables S1-S5 accordingly.

Rowan Trebilco

Response to Reviewer #2

This is a very intriguing piece of work. How to include of large, mobile, generalist consumers in food web theory is one of the most fascinating aspects of community ecology. This paper focuses on the size spectrum, and provides a new way to integrate unusually large and generalist predators into size-spectral theory. Basically, the author(s) demonstrate that the inclusion of these predators result in feasible top-heavy hourglass ecosystems, without invoking energetic subsidies, overestimation, or prey refuges, as done previously. They do so by means of a steady-state model using metabolic theory reasoning across trophic compartments, and illustrate their findings with examples from marine ecosystems (coral reefs mostly). But their model can be applicable to terrestrial systems. While I found the question fascinating, and I really like the generality of the approach taken, I have two major concerns with this manuscript that, unless they are solved, I cannot fully evaluate the validity and potential impact of this paper.

We would like to thank the reviewer for constructive comments that will improve the manuscript. Our responses to specific comments are detailed below.

First, large generalist predators and mega-consumers increase the total biomass and reduce the mean trophic level in an ecosystem- in particular, in marine ecosystems. The authors claim that size-spectral theory does not account for generalist predators or animals that feed lower in the food web than predicted from size alone. I do not see why that is the case. In equation 1, TE and PPMR can be calculated for any feeding structure by simply averaging effects. The $\frac{1}{4}$ exponent, however, is not clear to me whether it assumes feeding on a single resource pool (e.g., a single trophic level). Unless the author(s) explained more clearly what the basis and the gap in the theory comes from, it is very difficult to evaluate the novelty and potential impact of this paper in the area.

Currently, size-spectral theory does not account for LGPs and GSCs as detailed in a recent review (Blanchard et al 2017) which identified the ability to incorporate such trophic groups as a major next step in the development of size-spectral theory for marine ecosystems. Currently, size-spectral theory uses fairly limited prey ranges for predators and simply averaging does work to predict ecosystem structure. However, we show that simply averaging effects does not elucidate the same ecosystem structure when including LGPs and GSCs. We have added text to reflect this issue (Lines 70-74). This is a very nice addition and we are grateful for the insight.

Second, the author(s) state that the spatial scale over which size-spectra are calculated corresponds to the range of the largest or most mobile animal in the system. Also, that the existence of energetic subsidies for large and generalist predators is not well integrated into the size-spectral theory. With their toy model, the authors show that there is no need to claim the existence of prey refugia or energetic subsidies. This is novel and counter-intuitive. However, it remains to be proven that the spatial scale at which these large and generalist predators feed is the one contemplated in size-spectral theory, or in the model used here. Some of these predators have migration patterns that would require building size spectra at large spatial extents. The question that remains is: would current theory still work if size-spectra were built at the scale at which these large and generalist predators actually operate? If so, do we still need the theory developed here?

We agree completely with the questions posed by the reviewer. We state that the spatial scale over which size-spectra are calculated should be at the scale of the largest animal in the system. In applications of size-spectral theory, models are applied at the scales of a single reef up to the entire North Sea. Since size-spectra work across this wide range of scales, we believe size-spectral theory is applicable at these large spatial scales and our model is a step in the development of that theory to include the unique predation preferences of GSCs and LGPs. We show that size-spectra developed at smaller spatial scales require invoking energetic subsidies or prey refuges, but that at the largest scales, this cannot be the

case. *The ocean ecosystem can support very large abundances of large animals. We have modified the text to make this clear (Line 66-68).*

Minor points

- Manuscript is well written but the lack of subheadings makes it difficult to navigate and focus attention. I strongly suggest using key subheadings.

Done as suggested. We have added three subheadings (Size-spectral theory, Line 49; Feeding beyond PPMR, Line 84; Implications for ecosystems, Line 191)

- I found the Intro very repetitive- lines 1-74 can be shortened to half their current extent.

Since we have separated lines 1-74 into 2 sections now, we feel that the introduction is not repetitive. After line 46, we take a step back to familiarize the reader with size-spectral theory to highlight the limits of current theory. We believe this response to the comment about section headings above resolves this issue.

- Use of abundance and biomass as interchangeable terms? If by abundance, the author(s) refer to numerical abundance, predictions are very different. For example, Figure 1 only makes sense for biomass, not for numerical abundance. Please use only biomass throughout the manuscript.

We only reference biomass in all instances in the revised version

- Lines 14-16: empirical evidence for this should be cited.

Lines 14-16 refer to results of this manuscript. We have added citations for empirical observations to support our result (Line 16).

- Line 23: references needed

Done as suggested (Line 23).

- Figure 1: grey shading (based on the distributions of k for marine ecosystems)

Corrected as suggested.

- Line 48: Where equation 1 is derived from? This is the key theoretical construct of the manuscript, yet it seems to emerge from nowhere.

References cited to address this (Line 58).

Lines 150-152: This is a very interesting and convincing argument. However, the authors should do a better job in explaining the equations.

We believe that providing the reference for equation (1) to which this argument derives from satisfies this explanation.

- Line 161: should read “larger than that from lower...”

Corrected (Line 185).

- Line 197-199: references are needed here. It is not clear to me why this should be the case. Based on stoichiometric reasoning, animal-animal interactions should have larger efficiencies than herbivore-phyto interactions- hence efficiencies should increase with size.

We have added references accordingly. Barnes et al (2010) show that TE is expected to decrease with size likely due to the increase in metabolic energy allocated to foraging. We have added the reference and a sentence explaining this (Lines 229-233).

- Supplementary Table 1: please explain what abbreviations mean

Done as suggested.

Response to Reviewer #3

The authors present a model that predicts the size spectrum of a community that contains both large generalist predators (LGP) and gigantic secondary consumers (GSC), both of which are not well-predicted by current size spectrum theory. The model is a trophic compartment model that is based on metabolic theory and steady state assumptions. The authors show that their model predicts an hourglass-shaped size spectrum where for \leq fish size class, there is a decrease in biomass with increasing body size; above the fish size class there is an increase in biomass with increasing body size when LGPs and GSCs are incorporated into the model. This hourglass shape is energetically feasible and is supported by data collected from pristine communities - it appears that heavily anthropogenically altered communities may lose the top part of the hourglass, biasing observations towards a linear (negatively sloped) size spectra. I thought that this manuscript was well-written, appeared to be theoretically sound, and provided an important link connecting communities with GSCs and LGPs to the larger body of size spectra theory.

We thank the reviewer for the constructive comments especially the links to terrestrial ecosystems which will generalize our findings further. We have provided responses to specific comments below.

General comments

It is eluded to several times that human-influenced communities may be absent of GSCs and LGPs, such that anthropogenically altered systems will fit well with traditional size spectrum theory. Mentioned below is a reference that reports similar biomass skews in terrestrial African systems, which are more Pleistocene-like than other terrestrial mammal systems. Incorporating these terrestrial studies quantitatively into the analyses, or spending some time discussing similarities/differences of these systems with the presented marine systems may help to generalize the findings presented here. Moreover, I felt that the anthropogenic alteration angle was mentioned in the beginning of the manuscript, but not revisited satisfactorily towards the end of the paper.

We thank the reviewer for pointing us to these terrestrial linkages and have added discussion accordingly. We have also revisited the anthropogenic alteration angle again at the end of the paper (Lines 248-253, 256-258,260-261).

L113: This is also something that has been recently shown for African mammalian systems, where the biomass of megafaunal herbivores such as elephants appears to account for a large portion of mammalian biomass in the system - see Hempson et al. (Science, 2015). From Hempson: “Elephants dominate African herbivore biomass, often having biomasses equivalent to those of all other species combined.”, which seems relevant here.

We have added this citation as a bridge to terrestrial work as suggested (Line 258).

It seems like the trophic compartment model is minimalist in the sense that it works well to address the primary questions that the authors are tackling. It is also somewhat divorced from other trophic web models in that it does not consider individual species, but aggregates of individuals from many species within a given trophic compartment (if I understand the model correctly). I'm not suggesting that the authors adopt a different approach, as this one appears to work very well, however I think that drawing some links between the trophic compartment models here and species-explicit trophic web frameworks might be informative/interesting, and help to connect this work to some of the size-based food web frameworks that exist.

Great point, we have added some links to species-specific models as suggested. Specifically, we discuss how species-specific models generally do not show these patterns, but may underestimate production terms (Lines 211-216).

Of particular relevance is the paper by Rudolf Rohr (cited below), where they model food webs based on predator-prey mass ratios and have additional ‘latent traits’ that define non-body mass dictated relationships (e.g. baleen whales). I think that the impact of this contribution would be extended by finding the common ground between these two bodies of work, where one does not find much cross-pollination (surprisingly).

Again, great point; we have added a sentence or two to visit this issue accordingly. We have added most of the references below to better cast our findings within the body of knowledge on allometric relationships (Lines 2,8,94).

Some references in the food web world that deal heavily with allometric relationships:

-Brose, U., Williams, R. J., & Martinez, N. D. (2006). Allometric scaling enhances stability in complex food webs. *Ecology Letters*, 9(11), 1228–1236. <http://doi.org/10.1111/j.1461-0248.2006.00978.x>

-Petchey, O. L., Beckerman, A. P., Riede, J. O., & Warren, P. H. (2008). Size, foraging, and food web structure. *Proceedings of the National Academy of Sciences of the USA*, 105(11), 4191.

-Rohr, R. P., Scherer, H., Kehrl, P., Mazza, C., & Bersier, L.-F. (2010). Modeling food webs: Exploring unexplained structure using latent traits. *American Naturalist*, 176(2), 170–177. <http://doi.org/10.1086/653667>

-Woodward, G., Ebenman, B., Emmerson, M., Montoya, J. M., Olesen, J. M., Valido, A., & Warren, P. H. (2005). Body size in ecological networks. *Trends in Ecology and Evolution*, 20(7), 402–409.

Minor Comments

L42: consider stipulating what $PPMR > 1$ or < 1 means, so that readers will know for certain whether prey or predator mass is in the numerator. Depending on the method, predator-prey mass ratios are not always presented with one of the two always being in the numerator, which can make things confusing if the reader is used to a different standard.

Done as suggested. We only use Predator-Prey Mass Ratio which is predator mass/prey mass. We have added a sentence to clarify (Lines 50-54)

L50: This line is confusing to me. $\log(TE)$ will certainly be negative due to thermodynamics, but the log ratio will only be negative if the predator is larger than the prey. You say this in the sentence, but it is confusingly put. The relation doesn’t ‘require’ the log ratio to be negative because it can easily be positive, i.e. if the predator is smaller than the prey, which is often the case, particularly in terrestrial systems.

We have reworded this sentence to clarify (Lines 61-64).

Eq1: Consider mentioning the scaling law that determines the intercept - otherwise it just looks a bit random.

Done as suggested (Lines 61-62).

Fig 2: The red line in each subplot is a bit confusing... My eye wanted to match it to the red line in (a) where the points are also red. Consider using the same-color, but stippled or dotted, line to show the fit up to fish size, since the points are color coded from (a-d).

Done as suggested.

Fig. 3a is not referenced in the text.

Reference added (Line 138).

L233: is it specified that j is trophic level? I think it is, but I'm not sure. Now I see it is, but consider mentioning this when the subscript is first used.

Done as suggested (Lines 284-285).

Figures in general: consider putting more descriptive names on the axes rather than just the parameter. Then you don't have to dig through the paper to figure out what is being conveyed.

Done as suggested. We have added 'Individual Mass' and 'Biomass' to the x and y axes of Figs 2-4 respectively where appropriate.

Reviewers' comments:

Reviewer #1 (Remarks to the Author):

In my previous review, my assessment was that this manuscript has the potential to be a novel and important contribution, but more careful consideration of the assumptions and potential flaws of the proposed model was required, along with a more balanced and considered delivery. In revising the MS the authors improved their explanation of the model and in doing so addressed one of the major issues that I raised relating to the model assumptions (the issue of apparent "double dipping"). However most of the other issues I raised have only been partially addressed. I provide specific comments below that will hopefully help more fully address these issues.

- as highlighted in several of the papers cited in this MS (notably Blanchard et al 2017 and Andersen et al 2015), a diverse body of theory and models have been developed for size spectra in recent decades. Broadly, these fall into 2 categories - static "scaling" models and dynamical models. What the authors refer here as "size spectral theory" is essentially the first of these categories - static scaling models. Dynamical models are not considered, but these can treat feeding strategies much more flexibly. It would be good to modify wording to reflect this. For example, in the abstract reword the 3rd sentence to "... deviate from the simplest size-based predictions... ". Other use of "size spectral theory" should also be checked.

- on a related note, the terms "size spectrum" or "size spectra theory" are widely used in the literature; and I'd suggest sticking with this convention unless the authors can see a pressing need to instead use "spectral"

- as also noted by reviewer 2, it is inaccurate to say that size spectra models do not account for generalist predator behaviour. Indeed they assume extreme generalism from the perspective that all prey of appropriate size are equal, regardless of species. The important point is that static scaling size spectrum models assume relative size preference is constant across body sizes.

- The manuscript compares configurations of the compartment model: a base case, where the largest compartments are "large fish" and "apex predators"; is compared with scenarios where "large fish" are replaced by "LGPs" and/or "apex predators" are replaced by "GSCs". Why is the apex predator category included in the 1st model, but not the other 2? It seems strange to replace the apex predator class with the GSC class - wouldn't it instead make more sense to have both AP and GSC groups? This is particularly relevant for comparing the LGP+GSC model to other scenarios - because in this case you would expect the LGP group to be "released" from predation pressure in the absence of APs?

- one issue I raised in my previous review was that sensitivity to the number of compartments has not been considered. In the response, the authors note that they consider this in another manuscript, which is a fair response. However I still think its important to explicitly note in the discussion that the result may be sensitive to this, and can identify that this is an important avenue for future work.

- the point of my previous comment on minimum viable population sizes is that the densities reported can be used to calculate what total ocean area would be required to sustain a given population size in one of the compartments. How much ocean area is required to support minimum viable populations at the body sizes of LGPs and GSCs? How does this compare with the reef area at locations like Palmyra?

- abstract: add "at large body size" to 4th sentence to read "Here we show that generalist predatory behavior and lower trophic feeding at large body size..."

- lines 12-14: Overstated as written. Suggest rewording along the lines of "Here, using a simple model, we show how the inclusion of LGPs and GSCs could substantially increase total biomass and reduces mean trophic level"
- line 17: reword to "...and provide new perspectives on baselines..."
- line 45-46: suggest rewording to "... that feed well below the prey sizes and/or on a wider range of prey sizes than typically assumed in size spectra models and theory" as size spectra theory does not predict predict
- lines 50-59: I think a more nuanced treatment of PPMR is needed here, particularly to explain how it is defined for use in the macroecological size spectrum model (line 60). In this context it is defined as the mean (for the whole community) ratio of mass at TL n: mass at TL n-1.
- line 69: as noted above, its inaccurate to say "size-spectral theory currently does not account for generalist predators". Reword.
- lines 76-78: I raised a concern with the wording of this part of the MS in my previous review and while there has been a minor rewording, my concern remains. It under-values previous work and over-sells this MS to say "none of the aforementioned explanations (overestimation, energetic subsidies) provides a theoretical bridge between these conflicting findings". As a solution, I suggest deleting the first 2 sentences of this paragraph, and just start with "Here we show..."
- line 81-82: I'm not sure that its accurate to say "This finding provides a new unifying concept and explanation" - as the potential importance of "feeding down food chains" has been noted numerous times previously in the literature. I suggest toning this down.
- lines 85-87: as noted in my previous review, its not clear that there is compelling evidence in the literature that carcarhinid sharks feed on a broader range of prey, relative to their body size, than do smaller teleost fish. The analogy with bears (which are omnivores) is a big stretch. See also my comments relating to Table S1. The case for LGPs having a fundamentally different feeding mode to that at smaller body sizes needs to be strengthened.
- lines 96-105: I still find a few elements of the explanations of LGPs and GSCs here a bit confusing. The compartment model presented here has 6 trophic compartments - or trophic levels. As far as I can tell, LGPs are supposed to fall in TL 5, while true apex predators (things like large adult white, tiger, hammerhead and dusky sharks) should be in TL 6. We'd expect GSCs to be in TL 6 as well based on their size, but their diets are more akin to compartment 3. But the terminology and numbering from lines 96-105 doesn't line up with this - LGPs are referred to as "apex" predators, and it says we would expect GSCs to occupy TL 5 rather than 6 based on their body size.
- Line 192: reword to "traditional biomass pyramids should not always be expected" i.e. add the word "always". As written this is a big over-generalisation.
- line 211-216: This paragraph needs some work. The meaning of the sentence "However, the ability of these models to represent fine-scale predator-prey interactions may underestimate their predictive power" is not clear. Nor is it clear why production at trophic levels could be 10-100 times higher as stated.

- line 235-247: this is a nice addition but something I raised in my previous review and is still not dealt with here is how big does the ecosystem have to be to sustain minimum viable populations of the largest consumers? If this is bigger than the local systems then subsidies will still be necessary.
- line 254-256: reword along the lines of "Our results indicate that top-heavy trophic structure may be possible...". As written it is inappropriately definitive given the model/hypothetical nature of the approach.
- line 262-264: This should be reworded along the lines of "this study provides an alternative complimentary approach for generating baseline expectations of what ecosystem structure would be without extraction and other impacts". This is not the first study to provide information on baselines, nor does it have fewer assumptions than previously published approaches.
- line 264-267: the sentence "Our results also unify..." is an overstatement, and I suggest deleting it.
- line 337: As the conclusions of this article are based on custom computer code, the code availability statement needs to specify how the code can be accessed. Just providing code for your figures does not make this reproducible. If you are going to post the model code to github as indicated in your response letter, you should provide the link here,.
- figure 5: this figure needs axis labelling and scales.
- line 380: author initials are incorrect and 2nd author name missing for Tucker (& Rogers) 2014
- Table S1: where did the values here come from? The prey weights provided seem much more likely to be reflective of minimum prey sizes than means - especially for sharks. It seems very unlikely that the *average* prey weight for 100 kg copper and 200 kg bull sharks are 80 g and 100 g respectively, hence I suspect the PPMRs in this table are substantial over-estimates.

Rowan Trebilco

Reviewer #2 (Remarks to the Author):

Review of "A unifying theory for top-heavy ecosystem structure in the ocean". 2nd round.

My previous concerns about this paper relied on the validity and potential impact of the paper. The authors have responded to my two major concerns convincingly, and I find the modified paper very strong and novel. Firstly, the gap of the size-spectral theory that this paper tries to fill is now clearly identified. Secondly, my concern about the scale of the current model is also solved. In addition, I found the paper much easier and attractive to read given the current subheadings. My only minor concern is the assumption of trophic efficiencies decreasing with size (lines 229-233). In freshwater benthic systems, for example, this is not the case. I guess this holds only if animal-animal interactions are considered (similar stoichiometric imbalances across trophic levels and body sizes), but the assumption is not valid if primary producer-herbivore interactions are contemplated along with predator-prey interactions.

Reviewer #3 (Remarks to the Author):

I have reread the manuscript and commend the authors on their edits. I think that the submission is a

worthy contribution to Nature Communications, and nearly all of my issues/concerns were satisfied in the current draft.

I noticed only one small grammatical error:
L159 - as a function [of]

Suggestion: Figure 3 - it might look cleaner to put the legend in the lower right of panel b

Reviewer #1 (Remarks to the Author):

In my previous review, my assessment was that this manuscript has the potential to be a novel and important contribution, but more careful consideration of the assumptions and potential flaws of the proposed model was required, along with a more balanced and considered delivery. In revising the MS the authors improved their explanation of the model and in doing so addressed one of the major issues that I raised relating to the model assumptions (the issue of apparent "double dipping"). However most of the other issues I raised have only been partially addressed. I provide specific comments below that will hopefully help more fully address these issues.

We thank Reviewer #1 for the detailed comments that have greatly improved the manuscript.

- as highlighted in several of the papers cited in this MS (notably Blanchard et al 2017 and Andersen et al 2015), a diverse body of theory and models have been developed for size spectra in recent decades. Broadly, these fall into 2 categories - static "scaling" models and dynamical models. What the authors refer here as "size spectral theory" is essentially the first of these categories - static scaling models. Dynamical models are not considered, but these can treat feeding strategies much more flexibly. It would be good to modify wording to reflect this. For example, in the abstract reword the 3rd sentence to "... deviate from the simplest size-based predictions... ". Other use of "size spectral theory" should also be checked.

Done as suggested. We added 'simplest' to the abstract and changed 'size-spectral' to 'size spectra' throughout.

- on a related note, the terms "size spectrum" or "size spectra theory" are widely used in the literature; and I'd suggest sticking with this convention unless the authors can see a pressing need to instead use "spectral"

Done as suggested. We changed 'size-spectral' to 'size spectra' throughout.

- as also noted by reviewer 2, it is inaccurate to say that size spectra models do not account for generalist predator behavior. Indeed they assume extreme generalism from the perspective that all prey of appropriate size are equal, regardless of species. The important point is that static scaling size spectrum models assume relative size preference is constant across body sizes.

Changed to more accurately reflect size spectra model assumptions (Lines 46-47). Reviewer 2 now seems fine with the wording so no significant change was made.

- The manuscript compares configurations of the compartment model: a base case, where the largest compartments are "large fish" and "apex predators"; is compared with scenarios where "large fish" are replaced by "LGPs" and/or "apex predators" are replaced by "GSCs". Why is the apex predator category included in the 1st model, but not the other 2? It seems strange to replace the apex predator class with the GSC class - wouldn't it instead make more sense to have both AP and GSC groups? This is particularly relevant for comparing the LGP+GSC model to other scenarios - because in this case you would expect the LGP group to be "released" from predation pressure in the absence of APs?

Apex predators are reclassified as Large Generalist Predators in the models. So that the LGP+GSC and the LGP only (which contains an additional predator above LGPs as an AP) is included. Since the models assume steady-state, there is no release effect, this would only occur in a dynamic model. Further, we specifically wanted to keep the number of compartments in the models the same, so some of this rearrangement was needed to fit this constraint. We have conducted models as suggested with LGPs,

GSCs, and Apex Predators (Supplementary Figure 4 and Table 6). There are no significant effects on the results. We added text to address this issue (Lines 164-168).

- one issue I raised in my previous review was that sensitivity to the number of compartments has not been considered. In the response, the authors note that they consider this in another manuscript, which is a fair response. However I still think its important to explicitly note in the discussion that the result may be sensitive to this, and can identify that this is an important avenue for future work.

Done as suggested. We also now include an analysis of the effects of from 3-10 trophic compartments using randomized food webs (only constraint is that top trophic compartment feeds across range of prey) and trophic efficiencies (drawn from normal distribution as $TE = 0.101 \pm 0.058$) (see Lines 160-164 and new Supplementary Figure 4).

- the point of my previous comment on minimum viable population sizes is that the densities reported can be used to calculate what total ocean area would be required to sustain a given population size in one of the compartments. How much ocean is required to support minimum viable populations at the body sizes of LGPs and GSCs? How does this compare with the reef area at locations like Palmyra?

We added text that discusses the minimum productivity required to address this issue in the previous revision as suggested by Reviewer #2. We chose this analysis over the minimum area for a viable population because minimum viable population sizes are highly uncertain. In response to this comment, we have now added an analysis of area required for a minimum viable population assuming the population of grey reef sharks is viable, or that the observed densities represent viable populations (Lines 253-261).

- abstract: add "at large body size" to 4th sentence to read "Here we show that generalist predatory behavior and lower trophic feeding at large body size..."

Done as suggested (in Abstract).

- lines 12-14: Overstated as written. Suggest rewording along the lines of "Here, using a simple model, we show how the inclusion of LGPs and GSCs could substantially increase total biomass and reduces mean trophic level"

Done as suggested (Lines 12-14).

- line 17: reword to "...and provide new perspectives on baselines..."

Done as suggested (Line 17).

- line 45-46: suggest rewording to "... that feed well below the prey sizes and/or on a wider range of prey sizes than typically assumed in size spectra models and theory" as size spectra theory does not predict

Done as suggested (Lines 44-45).

- lines 50-59: I think a more nuanced treatment of PPMR is needed here, particularly to explain how it is defined for use in the macroecological size spectrum model (line 60). In this context it is defined as the mean (for the whole community) ratio of mass at TL n: mass at TL n-1.

Done as suggested (Lines 53-55).

- line 69: as noted above, its inaccurate to say "size-spectral theory currently does not account for generalist predators". Reword

Done as suggested (Lines 71-72).

- lines 76-78: I raised a concern with the wording of this part of the MS in my previous review and while there has been a minor rewording, my concern remains. It under-values previous work and over-sells this MS to say "none of the aforementioned explanations (overestimation, energetic subsidies) provides a theoretical bridge between these conflicting findings". As a solution, I suggest deleting the first 2 sentences of this paragraph, and just start with "Here we show..."

Done as suggested (Line 78).

- line 81-82: I'm not sure that its accurate to say "This finding provides a new unifying concept and explanation" - as the potential importance of "feeding down food chains" has been noted numerous times previously in the literature. I suggest toning this down.

Done as suggested. We have reworded to say "This finding provides an alternate explanation for the empirical observation of top-heavy food webs, and can provide predictions of when and where these ecosystem structures may occur." (Lines 81-83).

- lines 85-87: as noted in my previous review, its not clear that there is compelling evidence in the literature that carcarhinid sharks feed on a broader range of prey, relative to their body size, than do smaller teleost fish. The analogy with bears (which are omnivores) is a big stretch. See also my comments relating to Table S1. The case for LGPs having a fundamentally different feeding mode to that at smaller body sizes needs to be strengthened.

Done as suggested (Lines 346-348, Supplementary Table 1). We added an analysis of Cortes et al 1999 and the related paper Snelson et al (1984). This increased prey size to 200 g for a large bull shark. We agree that bears might be a big stretch, but theoretically similar in that they feed over a large range of prey sizes. Since other reviewers did not see this as an issue we decided to leave as is. We base part of our analysis on results from Lucifora et al (2009) Figure 3 (provided below) demonstrating that large sharks continue to feed on very small prey. The mean prey size changes, but the median does not. Our results suggest if a large predator derives even a very small amount of energy from small prey (lower trophic level than predicted by size spectra) then the biomass can be significantly higher for a given level of production (Lines 182-186). We have added a description of how we classified LGPs and GSCs in the methods (Lines 355-362).

Fig. 3 Quantile regressions of prey weight and copper shark, *Cararhinus brachyurus*, total length. The dotted, dashed and solid lines are 10, 50 and 90% quantile regressions used to estimate changes with shark length in minimum, median and maximum prey mass, respectively. Only the regression for maximum prey mass is significant ($P < 0.005$). Dots: prey consumed sectioned, open circles: prey consumed whole

- lines 96-105: I still find a few elements of the explanations of LGPs and GSCs here a bit confusing. The compartment model presented here has 6 trophic compartments - or trophic levels. As far as I can tell,

LGPs are supposed to fall in TL 5, while true apex predators (things like large adult white, tiger, hammerhead and dusky sharks) should be in TL 6. We'd expect GSCs to be in TL 6 as well based on their size, but their diets are more akin to compartment 3. But the terminology and numbering from lines 96-105 doesn't line up with this - LGPs are referred to as "apex" predators, and it says we would expect GSCs to occupy TL 5 rather than 6 based on their body size.

Corrected as suggested (Lines 96-105). In our model, LGPs are considered apex predators since nothing consumes them, we call them LGPs to distinguish that we are accounting for the fact that many of these animals also feed on smaller prey. One conclusion is that it does not take much of the smaller prey to have an effect (<5% of diet). We have added a model run with a true apex predator showing our results do not change significantly (Supplementary Table 6 and Figure 5).

- Line 192: reword to "traditional biomass pyramids should not always be expected" i.e. add the word "always". As written this is a big over-generalization.

Done as suggested (Line 196).

- line 211-216: This paragraph needs some work. The meaning of the sentence "However, the ability of these models to represent fine-scale predator-prey interactions may underestimate their predictive power" is not clear. Nor is it clear why production at trophic levels could be 10-100 times higher as stated.

Done as suggested. We have added explanations related to the fine-scale aggregation of prey leading to higher than expected production. In addition, we have added a discussion of how underestimating carrying capacity will directly limit these models to traditional biomass pyramids. Our results suggest carrying capacity may be greatly underestimated (Lines 215-224).

- line 235-247: this is a nice addition but something I raised in my previous review and is still not dealt with here is how big does the ecosystem have to be to sustain minimum viable populations of the largest consumers? If this is bigger than the local systems then subsidies will still be necessary.

Done as suggested. The issue with this analysis is that it may be impossible to really know what a minimum viable population is. We therefore included this analysis assuming that the current population of sharks is viable. In this case, the area for Palmyra can support these predators (Lines 253-263).

- line 254-256: reword along the lines of "Our results indicate that top-heavy trophic structure may be possible...". As written it is inappropriately definitive given the model/hypothetical nature of the approach.

Done as suggested (Line 270).

- line 262-264: This should be reworded along the lines of "this study provides an alternative complimentary approach for generating baseline expectations of what ecosystem structure would be without extraction and other impacts". This is not the first study to provide information on baselines, nor does it have fewer assumptions than previously published approaches.

We agree. Changed as suggested (Lines 278-279).

- line 264-267: the sentence "Our results also unify..." is an overstatement, and I suggest deleting it.

Done as suggested.

- line 337: As the conclusions of this article are based on custom computer code, the code availability statement needs to specify how the code can be accessed. Just providing code for your figures does not make this reproducible. If you are going to post the model code to github as indicated in your response letter, you should provide the link here.

We will add the link once the manuscript is accepted accordingly.

- figure 5: this figure needs axis labelling and scales.

We have added a scale bar and axis labels for each trophic level as suggested.

- line 380: author initials are incorrect and 2nd author name missing for Tucker (& Rogers) 2014

Corrected.

- Table S1: where did the values here come from? The prey weights provided seem much more likely to be reflective of minimum prey sizes than means - especially for sharks. It seems very unlikely that the *average* prey weight for 100 kg copper and 200 kg bull sharks are 80 g and 100 g respectively, hence I suspect the PPMRs in this table are substantial over-estimates.

These data come directly from two published papers (Snelson et al 1984 and Lucifora et al 2009) that show the median prey size does not change across the size range of these shark species. While average mass is much higher, the median is much lower. We argue that the median is a more appropriate representation of prey size and computation of the PPMR. We recalculated the prey mass for bull sharks using Snelson et al (1984) and found the average prey size to be 200 g. We updated Table 2 accordingly.

Rowan Trebilco

Reviewer #2 (Remarks to the Author):

Review of “A unifying theory for top-heavy ecosystem structure in the ocean”. 2nd round.

My previous concerns about this paper relied on the validity and potential impact of the paper. The authors have responded to my two major concerns convincingly, and I find the modified paper very strong and novel. Firstly, the gap of the size-spectral theory that this paper tries to fill is now clearly identified. Secondly, my concern about the scale of the current model is also solved. In addition, I found the paper much easier and attractive to read given the current subheadings. My only minor concern is the assumption of trophic efficiencies decreasing with size (lines 229-233). In freshwater benthic systems, for example, this is not the case. I guess this holds only if animal-animal interactions are considered (similar stoichiometric imbalances across trophic levels and body sizes), but the assumption is not valid if primary producer-herbivore interactions are contemplated along with predator-prey interactions.

The idea that trophic efficiency decreases with size comes from meta-analyses of marine systems only and as the reviewer suggests, does not hold for many other systems. We have changed the text accordingly to just refer to TE in marine systems (Lines 237).

Reviewer #3 (Remarks to the Author):

I have reread the manuscript and commend the authors on their edits. I think that the submission is a worthy contribution to Nature Communications, and nearly all of my issues/concerns were satisfied in the current draft.

We thank reviewer #3 for their contributions in improving this manuscript.

I noticed only one small grammatical error: L159 - as a function [of]

Corrected (Line 156).

Suggestion: Figure 3 - it might look cleaner to put the legend in the lower right of panel b

Done as suggested.

REVIEWERS' COMMENTS:

Reviewer #1 (Remarks to the Author):

I have reread the manuscript, and the responses to my last round of comments. I commend the authors on their edits. I think this will be a very valuable contribution to the literature and look forward to seeing it published. Rowan Trebilco